Review

 

**Subject Area:**
cellular biology/molecular biology

muscle stem cell, satellite cell, quiescence, activation, exercise, ageing

**Author for correspondence:**
Michael A. Rudnicki
e-mail: mrudnicki@ohri.ca

†Both authors contributed equally to this study.

# Satellite cells in ageing: use it or lose it

William Chen[1,2,†], David Datzkiw[1,2,†] and Michael A. Rudnicki[1,2]

[1]Sprott Centre for Stem Cell Research, Regenerative Medicine Program, Ottawa Hospital Research Institute, Ottawa, Ontario, Canada K1H 8L6
[2]Department of Cellular and Molecular Medicine, Faculty of Medicine, University of Ottawa, Ottawa, Ontario, Canada K1H 8M5

MAR, 0000-0002-3866-5249

Individuals that maintain healthy skeletal tissue tend to live healthier, happier lives as proper muscle function enables maintenance of independence and actuation of autonomy. The onset of skeletal muscle decline begins around the age of 30, and muscle atrophy is associated with a number of serious morbidities and mortalities. Satellite cells are responsible for regeneration of skeletal muscle and enter a reversible non-dividing state of quiescence under homeostatic conditions. In response to injury, satellite cells are able to activate and re-enter the cell cycle, creating new cells to repair and create nascent muscle fibres while preserving a small population that can return to quiescence for future regenerative demands. However, in aged muscle, satellite cells that experience prolonged quiescence will undergo programmed cellular senescence, an irreversible non-dividing state that handicaps the regenerative capabilities of muscle. This review examines how periodic activation and cycling of satellite cells through exercise can mitigate senescence acquisition and myogenic decline.

## 1. Introduction

Skeletal muscle accounts for approximately 40% of the human body weight and its functions include maintaining posture, mobility, regulating body temperature, energy storage and soft tissue support. Given its many important functions, it is unsurprising that muscle strength is positively correlated with better quality of life and negatively correlated with all-cause mortality [1–4].

The relationship between muscle strength and morbidity is more apparent in geriatric populations where sarcopaenia, age-related muscle atrophy, can lead to falls, disability and mortality. For example, the majority of hip fractures occur within individuals over 65 years of age [5]. Atrophy of stabilizer muscles can lead to loss of balance, important in preventing falls. Diminishing grip strength reduces one's ability to catch oneself in the event of a fall, and in the event of an accident, there is less muscle mass to cushion upon impact. Together, the greater risk of falls combined with sarcopaenia and osteoporosis are why the elderly are at a heightened risk of hip fractures, one of the highest predictors of morbidity and mortality in geriatric populations, with 1 year mortalities ranging from 14% to 58% [5,6]. Overall muscle decline begins around the age of 30, with an average annual muscle mass loss of 3–5%, with a total loss of up to 30% within an individual's lifetime [7–9].

Fortunately, skeletal muscle has remarkable regenerative capabilities, and age-related muscle loss can be drastically attenuated with exercise. On a physiological level, regular exercise has been shown to increase vascularization and perfusion, important for waste removal; muscle efficiency and greater recruitment of myofibres; flexibility and muscle tension; and bone strength and density, required to support muscles [10–12].

On the cellular level, exercise has been shown to increase mitochondrial numbers and quality, enhance innervation by neuromuscular junctions, stimulate supporting cells in the interstitial area of muscle (i.e. fibroapidogenic progenitors,

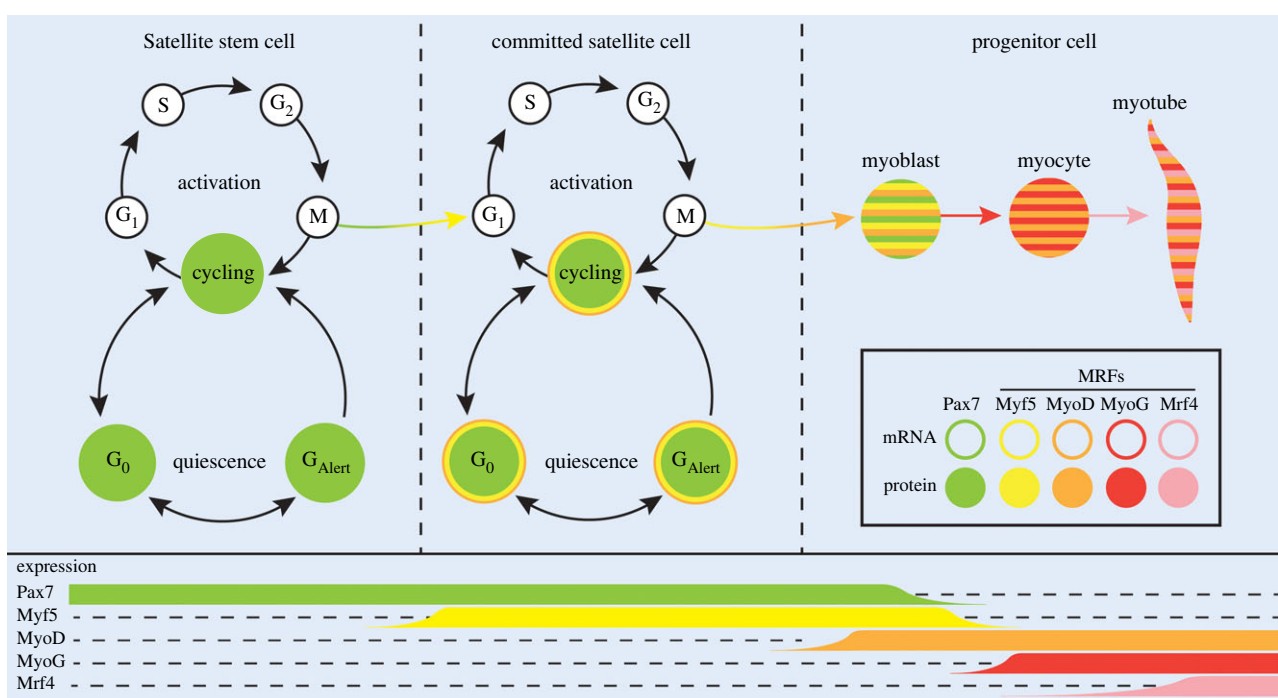

**Figure 1.** Satellite cell quiescence versus activation. Satellite stem (Pax7$^+$:Myf5$^-$) and committed cells (Pax7$^+$:Myf5$^+$) are able to exit the cell cycle and arrest in a reversible quiescent $G_0$ state. Satellite stem and committed cells are primed for rapid cell cycle reentry by transitioning to a second reversible quiescent state called $G_{Alert}$. Committed satellite cells in $G_{Alert}$ begin transcribing downstream MRFs *Myf5* and *MyoD*. Satellite cells in $G_{Alert}$ are primed to quickly respond to regenerative demands by entering the cell cycle faster than $G_0$ quiescent satellite cells. Active progenitor cells initiate the terminal differentiation programme to upregulate MyoG and Mrf4 in myoblasts and myocytes before fusion into myotubes.

FAPs), and stimulate satellite cells to activate for muscle repair and growth [13–16].

In this review, we address the role of quiescence in muscle regeneration, the mechanisms that prevent senescence in quiescent muscle stem cells, and how exercise contributes to long-term satellite cell viability at a cellular and molecular level.

## 2. Muscle regeneration and the satellite cell niche

Molecular and genetic studies in mice have greatly elucidated our understanding of the mechanisms that govern satellite cell function during the growth and regeneration of skeletal muscle [17–20]. Satellite cells, named after their satellite position on the myofibre, are a heterogeneous population of stem cells and committed cells that are required for muscle repair. Satellite cells highly express the transcription factor *Pax7* and sequentially express myogenic regulatory factors (MRFs) (*Myf5*, *MyoD*, *MyoG*, *Mrf4*) as they proliferate and then differentiate [21–24]. Pax7+ satellite stem cells mark their commitment to the myogenic lineage as committed myogenic cells by upregulating Myf5. The onset of Myf5 and MyoD protein expression marks the transition into myoblasts. Myogenin induction marks the entry of myoblasts into the terminal differentiation programme and subsequent fusion with other differentiating cells or with existing myofibres (figure 1) [25–27].

Approximately 10% of the satellite cell population appear to be a long-term self-renewing stem cell population expressing *Pax7* but not *Myf5*. The remaining 90% of the satellite cell population are committed cells that have expressed Myf5 and appear to form a short-term self-renewing population. Transplantation and serial injury experiments have demonstrated the satellite stem cell population to be capable of replenishing

the stem cell population in addition to re-establishing the pool of committed satellite cells [22].

Central to a satellite cell's behaviour is the immediate microenvironment that envelops the cell with structural proteins, and the various signalling factors from other nearby cell types. The nearest structures in proximity to a quiescent satellite cell include matrices of the fibrous sheath of basal lamina closest to the apical side of the satellite cell, and the plasma membrane of the myofibre at the basal side of the satellite cell. Satellite cells are often closely associated with capillary endothelial cells, as muscle tends to be a highly vascularized tissue [28]. Other cell types of the local milieu including fibrocyte/adipocyte progenitors (FAPS), pericytes and motor neurons can be found in the muscle interstitium [17]. Under homeostatic conditions, the satellite cell niche provides cues to actively maintain quiescence signalling pathways while suppressing activation pathways [29,30]. Upon injury, these cues activate satellite cells to prepare for the anticipated demands of regeneration [31]. Inflammatory cytokines are also released to provide chemotaxic cues for leucocytes, such as the macrophages required to clear debris and remove toxic waste [32–36]. Preservation of the stem cell pool is critical for maintaining regeneration potential, and thus the niche also provides cues for satellite cells to return to quiescence when regeneration is complete.

While the young niche favours stem cell quiescence, the aged niche provides cues that promote activation and differentiation [17,29,37–47]. Importantly, aged satellite cells demonstrate intrinsic defects that exposure to young niche environments cannot overcome, favouring differentiation and senescence rather than quiescence [38,48–52]. The aged niche and age-related stem cell defects lead to the typical fibro-adipogenic phenotype seen in aged and dystrophic muscle. These findings suggest that prolonged, chronic activation leads

royalsocietypublishing.org/journal/rsob    Open Biol. **10**: 200048

to stem cell exhaustion, and that maintenance of stem cell quiescence is paramount to maintaining a healthy stem cell population throughout the ageing process.

## 3. Satellite cells prefer quiescence

Homeostatic quiescence and activation is critical for establishing short- and long-term regenerative potential of the muscle by activating satellite cells and creating new progenitors for muscle repair, while preserving a small population of stem and committed satellite cells that can enter quiescence for the future regenerative demands.

In response to injury or inflammation, satellite cells will activate and enter the cell cycle to begin dividing and creating new cells for regeneration. Depending on the degree of regeneration required, satellite stem cells (Pax7$^+$/Myf5$^-$) will change their preference for asymmetric self-renewal or symmetric expansion of the stem cell pool. Under normal homeostatic turnover or minor damage, satellite cells preferentially undergo asymmetric division that will produce one stem cell (Pax7$^+$/Myf5$^-$) and one committed cell (Pax7$^+$/Myf5$^+$), which can then become a progenitor and undergo successive rounds of division [17,22,53]. Conversely, under more severe circumstances where damage is greater and demand for regenerative cells higher, satellite stem cells initially undergo symmetric division in order to rapidly expand the stem cell pool to facilitate large-scale generation of progenitors through asymmetric divisions [18,22,53,54].

Under homeostatic conditions, or following injury when active muscle repair is no longer required, satellite cells must be able to exit the cell cycle and enter a reversible state of quiescence to preserve and maintain long-term regeneration potential of its resident muscle [55,56]. Multiple studies have demonstrated that the loss of quiescence and the ability to exit and re-enter quiescence leads to a loss of 'stemness', homeostatic decline of the satellite cell population, and eventual inability to regenerate damaged muscle [57–59].

Transcriptomics and proteomic analysis of quiescent satellite cells have revealed a number of transmembrane receptors that are highly expressed within this population, including M-/N- cadherin junctions, Sprouty1, integrinβ1, frizzled receptors, OSMβ, CalcR and Notch receptors [60,61]. These receptors are part of a number of pleiotropic signalling cascades involved in actively maintaining quiescence by preserving cell/niche polarity, inhibition of cell motility and migration, regulating cytoskeletal organization, inhibiting gene of transcription/translation, inhibition of cell proliferation, and repression of activation-driving signalling axes such as ERK and hippo pathways [57,62–71]. Quiescent cells also maintain their $G_0$ state by regulating chromatin compaction and slowing protein synthesis [72,73].

In response to inflammation and injury, satellite cells can enter a second reversible quiescent state before activation called $G_{Alert}$ [74]. Committed satellite cells in $G_{Alert}$ are primed to express downstream targets of the myogenic lineage and withhold protein expression through post-transcriptional and post-translational modifications [75,76]. Satellite cells in $G_{Alert}$ enter the cell cycle at least twice as fast as satellite cells in $G_0$, facilitating a rapid response to injury. These findings suggest a hierarchical cascade model of quiescence and satellite cell recruitment during regeneration wherein quiescent $G_0$ and $G_{Alert}$ populations are present at each stage of the myogenic lineage (figure 1).

Unlike the serious injury models and techniques (i.e. cardiotoxin, BaCl$_2$, freeze and crush injuries) routinely performed on animal models to investigate muscle regeneration [77], humans do not typically experience such traumatic injuries during their lifetime. Thus, the real-world demand on quiescent satellite cells to activate and regenerate is comparatively lower, involving mainly homeostatic turnover and the response to exercise.

Given the impressive regenerative properties of skeletal muscle and lower demands of the modern day lifestyle, the reasons behind early adult onset of muscle decline remain elusive [7,8]. Recent findings on satellite cell quiescence and exercise in ageing populations provide a new paradigm on how to prevent age-related muscle regenerative decline.

## 4. Quiescence is a double-edged sword

Reversible quiescence is necessary in maintaining long-term regenerative potential, as a subset of regenerative stem cells and progenitors need to be preserved to adequately respond to future demands. However, periods of prolonged satellite cell quiescence may negatively impact long-term viability, leading to impaired regenerative potential (figure 2$a$). Despite lower protein synthesis, metabolism and general cellular activity, quiescent satellite cells still require cellular processes to actively maintain quiescence. These processes generate cellular waste, organelle damage and mitochondrial ROS from energy production (figure 2$b$) [72–74,78]. Accumulation of these waste products can harm the long-term viability of quiescent satellite cells. Fortunately, quiescent satellite cells have a number of protective mechanisms to prevent a build-up of cell waste products, DNA damage and transition to senescence [51,79–83].

Quiescent satellite cells under prolonged periods of quiescence are at risk of proteotoxicity (figure 2$b$). During deep quiescence, accumulation of dysfunctional proteins and organelles that would otherwise be diluted through rounds of cell division can increase cytotoxicity leading to senescence and apoptosis [84–86]. To combat proteotoxicity, systems for cell product turnover are important for maintaining cell viability.

The ubiquitin–proteasome system is one of the primary systems involved in pruning the proteomic landscape. This system is critical for maintaining cell identity. For example, Pax7 expression is regulated by Nedd4, a ubiquitin-ligase that is expressed during both quiescence and activation [87]. Satellite cell-specific knockout studies of the proteasomal subunit *Rpt3* demonstrate the importance of the ubiquitin–proteasome system in maintaining quiescence [83]. Disruption of the proteasome through deletion of *Rpt3* significantly depletes the quiescent satellite cell population in resting muscle. Moreover, post-injury muscle regeneration is disrupted under *Rpt3* knockout conditions, with a significant decline in Pax7-expressing quiescent satellite cell numbers at both 5 and 15 days post injury.

Autophagy also plays a pivotal role in the cell's capacity to maintain cellular integrity through recycling of intracellular components, such as defective or surplus proteins and organelles [85,88]. For the purpose of this review, autophagy will be used in reference to macroautophagy, one of the main forms of autophagy that occur [89]. Autophagy is a critical cellular process in maintaining cell viability, and its disruption in satellite cells promotes muscle atrophy and mitochondrial

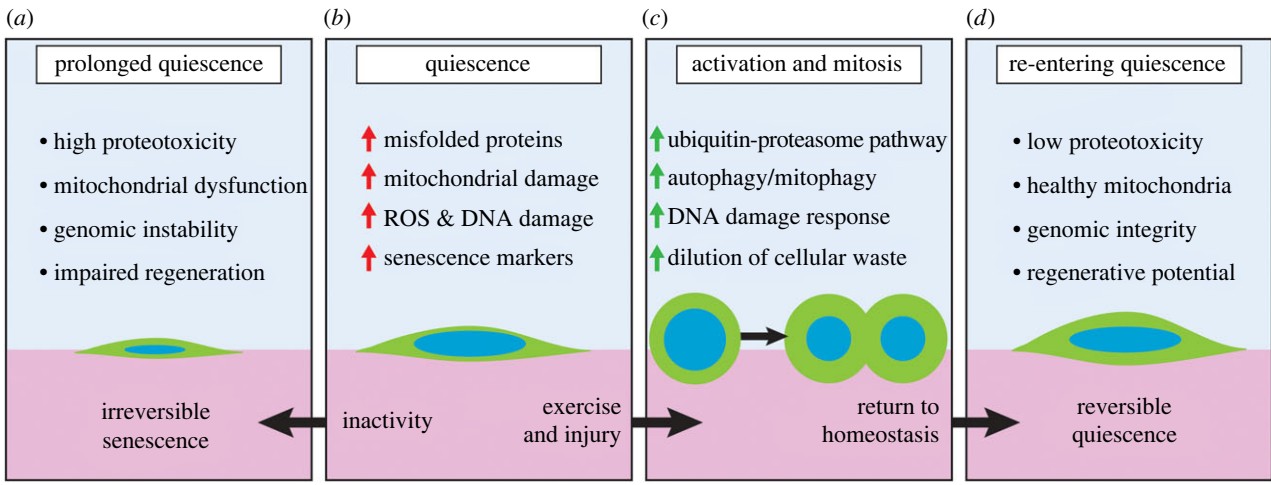

**Figure 2.** Healthy quiescence requires periodic activation. (*a*) Prolonged quiescence is met with detrimental effects, including high proteotoxicity, mitochondrial dysfunction, genomic instability and impaired regeneration. As satellite cells age, they are prone to prolonged states of quiescence during times of inactivity. (*b*) Quiescent satellite cells accumulate defects in the form of misfolded proteins, mitochondrial damage, reactive oxygen species (ROS) and DNA damage, and begin to express senescence markers. (*c*) Activation and mitosis of satellite cells induced through exercise or injury upregulates several processes that actively clear cellular waste, including the ubiquitin/proteasome pathway (UPP), autophagy/mitophagy and the DNA damage response (DDR). Passive dilution of cellular waste is also achieved through increased cytoplasm during mitosis. (*d*) During the return to homeostasis, satellite cells re-enter quiescence, exhibiting lower levels of proteotoxicity, healthy mitochondria, intact genomic integrity and a high regenerative potential.

dysfunction [90–93]. Proper autophagy function is required throughout the cell cycle, including active cycling and quiescence [51,94].

Pharmacological inhibition of autophagy and satellite cell-specific ablation of *Atg7*, an essential component of macroautophagy, demonstrate the requirement for a basal level of homeostatic autophagy in preventing quiescent satellite cells from transitioning into senescence [51]. Indeed, disrupted autophagy results in increased p62 and ubiquitin-positive aggregates, which mark damaged organelles and substrates for degradation respectively. Conversely, increased autophagy by *Atg7* overexpression is able to rescue the proliferative defect and reduce numbers of senescent satellite cells in geriatric mice [51].

All cells accumulate DNA damage from several intrinsic and extrinsic sources during ageing [95]. Satellite cells actively employ DNA damage responses (DDRs) as they activate and progress through the cell cycle towards differentiation [81,96]. Relatively little is known specifically concerning DDRs and satellite cell function. However, quiescent haematopoietic stem cells activate DDRs upon entry into the cell cycle to combat age-related DNA damage [97]. Presumably satellite cells upregulate DDRs in a similar fashion upon exiting quiescence and entering the cell cycle. The importance of a robust DDR in satellite cells can easily be expected, as this stem cell population must maintain in-tact genetic integrity to seed a lifetime of regeneration.

Quiescent satellite cells in $G_0$ display a detectable low level of energy metabolism and predominantly derive their energy needs from glycolysis, unlike most $G_1$-arrested cells that rely on oxidative phosphorylation [98–100]. Despite a preference for glycolysis in quiescent satellite cells, which does not rely on mitochondrial function [101], the ability to remove damaged mitochondria remains integral to maintaining cell viability. Mitophagy impairment results in increased reactive oxygen species (ROS), DNA damage and senescence markers that can be attenuated using pharmacological ROS inhibition [51]. Recent reports have shed light on this paradox and have demonstrated that fatty acid metabolism is also required to maintain an in-tact quiescent state [102,103], and

peroxisome-targeted inhibition of fatty acid oxidation lead to premature differentiation of myoblasts *in vitro* [103]. As satellite cells exit quiescence and enter the cell cycle, a metabolic switch to oxidative phosphorylation occurs, with an expected increase in mitochondrial density and recycling [99].

Unfortunately, these protective mechanisms in place are not enough to maintain long-term satellite cell viability and regenerative capabilities of muscle over the lifetime of an individual. Quiescent geriatric satellite cells eventually enter a pre-senescent state by de-repression of $p16^{INK4a}$ (Cdkn2α), which inhibits multiple quiescence-inducing pathways and increases DNA damage through a ROS-positive feedback loop [50,104]. Geriatric muscle satellite cells are not as capable and efficient in transitioning from $G_0$ quiescence to activation required for creating new progenitors and consequently are unable to keep up with muscle degradation. Indeed, muscle cross-sections of adult and geriatric mice compared to young mice exhibit signs of muscle decline: fibre atrophy, loss of innervation, and re-expression of embryonic myosin heavy chain and central nucleation [50,51,80,105–107].

It appears the autophagy and mitophagy pathways active during prolonged quiescence are not sufficient to prevent satellite cells from transitioning into senescence [82]. What, then, can we do to prevent this seemingly inevitable decline in homeostatic satellite cell regenerative function? We postulate that periodic activation and cycling of satellite cells is required to remove proteotoxic waste through cytoplasmic dilution and upregulation of autophagy to maintain long-term cell viability (figure 2*c,d*).

## 5. Exercise activates satellite cells

Exercise bestows several benefits to overall health, including improved bone strength, offset obesity [12], and improved cardiovascular [108], cognitive [109], and immune system functions [110]. One of the most apparent outcomes of exercise is improved function and health of skeletal muscle [11,111,112].

Here we will discuss the effect of exercise on the satellite cell population within skeletal muscle.

The muscle regeneration field has been focused on muscle formation during embryogenesis and regeneration following severe injury and disease progression. Less clear are the mechanisms regulating satellite cell activity during homeostasis, response to eccentric exercise and ageing. The satellite cell population maintains a level of functional heterogeneity, with subpopulations of satellite cells mounting differential responses depending on environmental stimuli [113,114]. Satellite cell lineage tracing experiments using multi-colour *Pax7CreERTM:R26RBrainbow2.1* mice reveal that clonal complexity is maintained during homeostatic ageing and undergoes clonal selection during severe muscle injury [54]. This suggests that the satellite cell response differs depending on different thresholds of stress. However, it is not clear as to what type and to what degree of response is potentiated following stress caused by various types of exercise.

Eccentric exercise enhances myofibre hypertrophy, and stimulates satellite cell activation and proliferation [115]. Aerobic endurance training stimulates satellite cell activation, and human strength training studies with both single and long-term training sessions result in a marked induction of satellite cell proliferation [116–126]. Proper satellite cell function is indeed responsible for exercise-induced muscle mass gain, as satellite cell depletion by irradiation results in the loss of expected hypertrophy following weight bearing exercises [127,128]. Indeed, satellite cell activation and fusion into existing myofibres are an integral part of myofibre growth and accumulation of myonuclei following exercise [129–131]. It is possible that impairment of satellite cell activation partly contributes to muscle atrophy observed in sedentary lifestyles [132–134] and during space flight [135] due to a loss of force-induced activation.

Paradoxically, activation of satellite cells through exercise enhances muscle function, yet the age-related dysfunction of satellite cells arises due to their heightened propensity to activate and differentiate. As discussed earlier, this is in part due to age-related changes intrinsic to satellite cells [38,48–52], and to the aged stem cell niche [17,29,37–47]. However, it appears that in contrast to the exhaustive chronic satellite cell activation associated with age, acute activation of satellite cells following exercise leads to a regenerative response that is met with a subsequent return to quiescence. Supporting this notion are studies demonstrating the benefits to overall muscle health in elderly people who perform resistance training, which stimulates satellite cell activation [115,136–142]. As such, exercise may drive significantly different satellite cell activation signalling compared to the response observed due to age-related defects. Besides signalling differences within the satellite cells, systemic factors may also be at play. For example, factors associated with exercise such as increased vascularization [143] may reduce chronic inflammation, allow for more efficient removal of cellular debris and waste build-up in the niche, and promote satellite cell function.

Current understanding of the factors that activate satellite cells during eccentric exercise is limited as most molecular studies describe satellite cells either in the non-stressed or severely injured state. It appears that satellite cell activation during exercise is mediated by elements released from the myofibre and interstitial cell populations. Signalling cascades that appear to be responsible for driving satellite cell activation in response to exercise induced muscle stress include IGF1, IL-6/JAK/STAT3, hippo and SIRT1 pathways [79,144–147].

IGF-1 is released from muscle fibres following mechanical strain and exercise, driving muscle hypertrophy [144,148]. Barton-Davis *et al.* observed that the hypertrophic effect of IGF1 was approximately halved in the absence of satellite cells in irradiation experiments [149]. Supporting these findings, other transient IGF1 overexpression studies in muscle report the accumulation of myonuclei, an event mediated by satellite cell fusion to existing myofibres [150]. Interestingly, a splice variant of IGF1 (IGF-1Eb), termed mechano-growth factor, is produced and released from muscle following weight-bearing exercise, and is thought to drive satellite cell proliferation [151]. However, this effect is contested with *in vitro* studies on C2C12 and human primary myoblast cultures failing to demonstrate any significant increase in proliferation following treatment with this IGF1 splice variant [152,153].

Interleukin 6 (IL-6) is a pro-inflammatory cytokine that is released from growing muscle [154]. IL-6 has been shown to be released following resistance training exercise, with concomitant satellite cell proliferation and STAT3 signalling activation [145]. STAT3 signalling is implicated in satellite cell progression through the myogenic lineage [145,155–157]. However, it should be noted that age-related chronic STAT3 activity inhibits satellite cell expansion and has been implicated in age-related satellite cell exhaustion [48,158].

Non-myogenic cells that reside within the interstitial spaces of muscle appear to also play a role in the IL-6 signalling axis post-exercise, namely FAPs and infiltrating eosinophils. FAPs provide functional support for satellite cells and influence their activation [155,159–163]. Following muscle injury, eosinophils infiltrate the interstitial space in close proximity to FAPs and satellite cells, and release interleukin 4 (IL-4) [160]. IL-4 release from eosinophils stimulates FAPs, which secrete IL-6 when activated, providing an additional source of this cytokine within a regenerating muscle [160,162,164]. Interestingly, eosinophils have been shown to secrete IL-4 when activated by Meteorin-like (Metrnl), a myokine that is secreted following exercise [165]. These findings demonstrate that satellite cell activation post-exercise is a multi-faceted phenomenon, with several cell types and factors working in concert to mediate a regenerative environment, similar to the response following severe injury.

Hippo signalling through YAP and Taz has been demonstrated to play a role in satellite cell activation and proliferation [166–168]. Interestingly, it is becoming more evident that mechanotransduction is a primary activator of the hippo pathway [169,170]. This is in line with a recent findings by Eliazer *et al.* that demonstrate Rho-dependent mechano-signalling to repress YAP during quiescence to prevent satellite cell activation [146]. These findings give rise to the idea that exercise-induced mechanotransduction may activate satellite cells through involvement of the hippo pathway. Additionally, physical activation by massage has been reported to activate and increase satellite cell numbers, suggesting that a sensitive mechano-sensing mechanism is involved in satellite cell activation under healthy conditions, which could potentially involve hippo signalling [171].

Lastly, SIRT1 is another signalling effector that has been implicated in satellite cell activation and is upregulated in muscle following exercise [147,172]. SIRT1 is among the family of sirtuin deacetylase proteins, which exert $NAD^+$-dependent activities and act as metabolic sensors [173]. SIRT1 is thought to upregulate autophagy as the cell exits quiescence to provide additional energy for activation and proliferation [79]. Deletion

of SIRT1 leads to a delay in satellite cell activation [79]. Indeed, satellite cell-specific SIRT1 knockout prevents muscle regeneration, and its overexpression improves muscle regeneration in older mice [174]. Since SIRT1 influences satellite cells through regulation of autophagy, it would be interesting to determine whether an age-related loss of SIRT1 expression is involved in the decline of satellite cell function linked to age-related autophagy dysfunction. Perhaps exercised-induced expression/activity of SIRT1 is capable of ameliorating dysfunctional autophagy in satellite cells.

Despite the well-established beneficial effect of resistance training, exercise alone does not prevent all age-related defects in skeletal muscle. Concomitant with sarcopaenia is a change in muscle composition within geriatric muscle: a shift of type 2 towards type 1 myofibres [175,176]. Even in cases of lifelong exercise, the shift from type 2 to type 1 myofibres persists [177] and the satellite cell population still diminishes to some degree over age [39,178–181]. Exercise-induced satellite cell activation also experiences some degree of impairment during ageing, as demonstrated by expression of myostatin within satellite cells, an inhibitor of satellite cell activation and self-renewal [178,181,182]. Regardless, lifelong resistance exercise should be pursued as the effects of sarcopaenia are greatly diminished when compared with sedentary lifestyles. As demonstrated in this review, the beneficial effects of exercise to muscle are derived from not only direct stimulation of muscle fibres, but also simultaneous activation of the muscle stem cell pool.

# 6. Concluding remarks

Higher quality of life and happiness is strongly associated with an individual's ability to exercise self-agency, and maintaining locomotion is a key aspect of this pursuit. Skeletal muscle health and exercise deserve greater attention in today's world, where a sedentary lifestyle is the norm, while obesity and cardiovascular disease are the leading causes of death. Although age-related muscle atrophy is the focus of many geriatric studies, it is clear that the decline in satellite cell number and function coincides with the gradual muscle loss that begins much earlier, at approximately 30 years of age.

Many additional mechanisms behind satellite cell function decline in ageing remain to be discovered. However, it appears to be multi-factorial and involve defects and misregulation of

cell maintenance, quiescence and activation pathways that prevent cell damage. It is evident that satellite cells are activated through various signalling pathways following exercise. Of particular relevance, cellular processes involved in the upkeep of cellular integrity (proteasome-mediated protein degradation, autophagy and DDRs) are upregulated during activation and proliferation. Moreover, satellite cells increase their cytoplasm during rounds of proliferation, which dilutes cellular waste and dysfunctional organelles between daughter cells. It is evident that waste and DNA damage accrued within periods of satellite cell quiescence are most effectively managed during periods of activation. Frequent activation of muscle satellite cells through exercise is a key process required to offset age-related waste/damage accumulation that leads to senescence. Physical inactivity becomes more detrimental with age as satellite cells accumulate higher levels of cellular waste and DNA damage during longer periods of deep quiescence.

Currently, our understanding of satellite cell behaviour in response to exercise is relatively limited compared with more severe models of injury. Future work in this field is needed to decipher the clonal dynamics of satellite cells following exercise. For example, is satellite cell replacement through asymmetric division sufficient to respond to exercise-induced stress, or is pre-expansion of the satellite cell pool required to mount a more robust response? Moreover, what is the threshold and range of exercise-induced mechanical stress that drives satellite cell activation, and perhaps different clonal dynamics? Are certain subpopulations of satellite cells more sensitive to mechanical stress within the normal range of exercise? Elucidating these mechanisms will reveal important information relevant to combating age-related muscle decline, as well as designing therapeutics to offset muscle atrophy that arises from sedentary lifestyle, space flight and disease.

Data accessibility. This article has no additional data.

Authors' contributions. W.C., D.D. and M.A.R. wrote the paper.

Competing interests. We declare we have no competing interests.

Funding. The studies from the laboratory of M.A.R. were carried out with support of grants from the Canadian Institutes for Health Research (FDN-148387), US National Institutes for Health (R01AR044031), Muscular Dystrophy Association, the Ontario Institute for Regenerative Medicine and the Stem Cell Network. M.A.R. holds the Canada Research Chair in Molecular Genetics. W.C. holds a University of Ottawa Brain and Mind Research Institute Éric Poulin Centre for Neuromuscular Disease Scholarship in Translation Research Award. D.D. holds a Frederick Banting and Charles Best Canada Graduate Scholarships–Doctoral Award (CGS-D).

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
