## [Reviewer comments · Open Biology]

Review History

RSOB-20-0048.R0 (Original submission)

Review form: Reviewer 1

Recommendation

Accept with minor revision (please list in comments)

Do you have any ethical concerns with this paper?

No

Comments to the Author

This is an interesting review, well written and with a compelling logical flow. It definitely is of interest to a large audience. However, there is space for improvement in a couple of specific areas.

The review is essentially divided in two sections, one that deals with aging and its effects on satellite cell function, somewhat neglecting the role of other components such as the inflammatory system and other tissue resident cells. This is not a problem but it should be acknowledged. The second part is focused on the effects of exercise. In this second part, the impact of aging is underdeveloped and almost tagged on as an afterthought. The review is still very well written and informative, but it does not seem to fulfill the expectations created by its title.

The only part of the manuscript that left me somewhat puzzled was the mention of the role of eosinophils, in which they are reported to express IL4/3... or IL3/4 in a different paragraph. IL3 and IL4 are quite different cytokines, and I am not aware of a hybrid molecule... in addition, eosinophils respond to and are attracted by IL3, but they are not, as far as I know, a major source of this cytokine. The authors may want to double check their sources.

Decision letter (RSOB-20-0048.R0)

16-Apr-2020

Dear Dr Rudnicki,

We are pleased to inform you that your manuscript RSOB-20-0048 entitled "Satellite Cells in Aging: Use It or Lose It" has been accepted by the Editor for publication in Open Biology. The reviewer has recommended publication, but also suggest some minor revisions to your manuscript. Therefore, we invite you to respond to the comments and revise your manuscript.

Please submit the revised version of your manuscript within 7 days. If you do not think you will be able to meet this date please let us know immediately and we can extend this deadline for you.

- 1) A text file of the manuscript (doc, txt, rtf or tex), including the references, tables (including captions) and figure captions. Please remove any tracked changes from the text before submission. PDF files are not an accepted format for the "Main Document".
- 2) A separate electronic file of each figure (tiff, EPS or print-quality PDF preferred). The format should be produced directly from original creation package, or original software format. Please note that PowerPoint files are not accepted.
- 3) Electronic supplementary material: this should be contained in a separate file from the main text and meet our ESM criteria (see <http://royalsocietypublishing.org/instructions-authors#question5>). All supplementary materials accompanying an accepted article will be treated as in their final form. They will be published alongside the paper on the journal website and posted on the online figshare repository. Files on figshare will be made available

approximately one week before the accompanying article so that the supplementary material can be attributed a unique DOI.

Online supplementary material will also carry the title and description provided during submission, so please ensure these are accurate and informative. Note that the Royal Society will not edit or typeset supplementary material and it will be hosted as provided. Please ensure that the supplementary material includes the paper details (authors, title, journal name, article DOI). Your article DOI will be 10.1098/rsob.2016[last 4 digits of e.g. 10.1098/rsob.20160049].

4) A media summary: a short non-technical summary (up to 100 words) of the key findings/importance of your manuscript. Please try to write in simple English, avoid jargon, explain the importance of the topic, outline the main implications and describe why this topic is newsworthy.

Images

Data-Sharing

It is a condition of publication that data supporting your paper are made available. Data should be made available either in the electronic supplementary material or through an appropriate repository. Details of how to access data should be included in your paper. Please see <http://royalsocietypublishing.org/site/authors/policy.xhtml#question6> for more details.

Data accessibility section

Sincerely,
The Open Biology Team
<mailto:openbiology@royalsociety.org>

Reviewer's Comments to Author:

Referee:

Comments to the Author(s)

This is an interesting review, well written and with a compelling logical flow. It definitely is of interest to a large audience. However, there is space for improvement in a couple of specific areas.

The review is essentially divided in two sections, one that deals with aging and its effects on satellite cell function, somewhat neglecting the role of other components such as the inflammatory system and other tissue resident cells. This is not a problem but it should be acknowledged. The second part is focused on the effects of exercise. In this second part, the impact of aging is underdeveloped and almost tagged on as an afterthought. The review is still

very well written and informative, but it does not seem to fulfill the expectations created by its title.

The only part of the manuscript that left me somewhat puzzled was the mention of the role of eosinophils, in which they are reported to express IL4/3... or IL3/4 in a different paragraph. IL3 and IL4 are quite different cytokines, and I am not aware of a hybrid molecule... in addition, eosinophils respond to and are attracted by IL3, but they are not, as far as I know, a major source of this cytokine. The authors may want to double check their sources.

Author's Response to Decision Letter for (RSOB-20-0048.R0)

See Appendix A.

Decision letter (RSOB-20-0048.R1)

23-Apr-2020

Dear Dr Rudnicki

We are pleased to inform you that your manuscript entitled "Satellite Cells in Aging: Use It or Lose It" has been accepted by the Editor for publication in Open Biology.

Sincerely,
The Open Biology Team
mailto:openbiology@royalsociety.org

Appendix A

Response to reviewer's comments

This is an interesting review, well written and with a compelling logical flow. It definitely is of interest to a large audience. However, there is space for improvement in a couple of specific areas.

The review is essentially divided in two sections, one that deals with aging and its effects on satellite cell function, somewhat neglecting the role of other components such as the inflammatory system and other tissue resident cells. This is not a problem but it should be acknowledged.

We would like to thank the reviewer's positive and constructive feedback. We agree that the niche and local milieu should be acknowledged in their roles during aging and exercise. We have updated our review to include these topics under section 2 now renamed to "Muscle regeneration and the satellite cell niche."

The second part is focused on the effects of exercise. In this second part, the impact of aging is underdeveloped and almost tagged on as an afterthought. The review is still very well written and informative, but it does not seem to fulfill the expectations created by its title.

We have reframed the article to highlight the important role of exercise in preventing long-term chronic satellite cell activation and quiescence, both extrinsically and intrinsically. We elaborate on the effect of exercise in aging to explicitly mention that exercise enhances muscle function. These changes should now better reflect the original intent the paper.

The only part of the manuscript that left me somewhat puzzled was the mention of the role of eosinophils, in which they are reported to express IL4/3... or IL3/4 in a different paragraph. IL3 and IL4 are quite different cytokines, and I am not aware of a hybrid molecule... in addition, eosinophils respond to and are attracted by IL3, but they are not, as far as I know, a major source of this cytokine. The authors may want to double check their sources.

Thank you for correcting this mistake. IL-13 was mistakenly replaced by IL-3. This has been fixed. Originally this section meant to describe the IL-4 / IL-13 signaling axis between FAPs and eosinophils. For clarity, IL-4 release is only mentioned as the source studies primarily focus on the role of this cytokine.